# Modeling Lung Carcinoids with Zebrafish Tumor Xenograft

**DOI:** 10.3390/ijms23158126

**Published:** 2022-07-23

**Authors:** Silvia Carra, Germano Gaudenzi, Alessandra Dicitore, Maria Celeste Cantone, Alice Plebani, Davide Saronni, Silvia Zappavigna, Michele Caraglia, Alessia Candeo, Andrea Bassi, Luca Persani, Giovanni Vitale

**Affiliations:** 1Laboratory of Endocrine and Metabolic Research, IRCCS, Istituto Auxologico Italiano, 20100 Milan, Italy; s.carra@auxologico.it (S.C.); luca.persani@unimi.it (L.P.); 2Laboratory of Geriatric and Oncologic Neuroendocrinology Research, IRCCS, Istituto Auxologico Italiano, 20100 Milan, Italy; g.gaudenzi@auxologico.it (G.G.); m.cantone@auxologico.it (M.C.C.); plebanialice94@gmail.com (A.P.); 3Department of Medical Biotechnology and Translational Medicine, University of Milan, 20100 Milan, Italy; alessandra.dicitore@unimi.it (A.D.); davide.saronni@unimi.it (D.S.); 4PhD Program in Experimental Medicine, University of Milan, 20100 Milan, Italy; 5Department of Precision Medicine, University of Campania “L. Vanvitelli”, 80138 Naples, Italy; silvia.zappavigna@unicampania.it (S.Z.); michele.caraglia@unicampania.it (M.C.); 6Laboratory of Molecular and Precision Oncology, Biogem scarl, 83031 Ariano Irpino, Italy; 7Department of Physics, Politecnico di Milano, 20133 Milan, Italy; alessia.candeo@polimi.it (A.C.); andrea1.bassi@polimi.it (A.B.)

**Keywords:** lung carcinoids, neuroendocrine tumors, typical carcinoid, atypical carcinoid, zebrafish, tumor xenograft, angiogenesis, metastasis, sulfatinib

## Abstract

Lung carcinoids are neuroendocrine tumors that comprise well-differentiated typical (TCs) and atypical carcinoids (ACs). Preclinical models are indispensable for cancer drug screening since current therapies for advanced carcinoids are not curative. We aimed to develop a novel in vivo model of lung carcinoids based on the xenograft of lung TC (NCI-H835, UMC-11, and NCI-H727) and AC (NCI-H720) cell lines and patient-derived cell cultures in *Tg(fli1a:EGFP)^y1^* zebrafish embryos. We exploited this platform to test the anti-tumor activity of sulfatinib. The tumorigenic potential of TC and AC implanted cells was evaluated by the quantification of tumor-induced angiogenesis and tumor cell migration as early as 24 h post-injection (hpi). The characterization of tumor-induced angiogenesis was performed in vivo and in real time, coupling the tumor xenograft with selective plane illumination microscopy on implanted zebrafish embryos. TC-implanted cells displayed a higher pro-angiogenic potential compared to AC cells, which inversely showed a relevant migratory behavior within 48 hpi. Sulfatinib inhibited tumor-induced angiogenesis, without affecting tumor cell spread in both TC and AC implanted embryos. In conclusion, zebrafish embryos implanted with TC and AC cells faithfully recapitulate the tumor behavior of human lung carcinoids and appear to be a promising platform for drug screening.

## 1. Introduction

Lung carcinoids are uncommon neuroendocrine tumors (NETs) accounting for 1–2% of all lung cancers. They include well-differentiated typical (TCs) and atypical carcinoids (ACs), which are considered low (G1) and intermediate (G2) grade NETs, respectively [1,2]. Although lung TCs and ACs have a generally more indolent behavior compared with other primary lung tumors, patients develop distant metastases in approximately 25–30% of cases. Despite the development of novel drugs in cancer research over the last decade, there are no curative therapies for advanced lung carcinoid tumors [3,4,5,6,7,8,9,10,11].

The key factor that has hindered clinical therapeutic progress is probably the lack of valid animal models. Although rodents represent the main animal model employed in cancer research, their use in the field of lung NETs is limited. Tumor xenograft murine models, particularly patient-derived xenografts (PDXs), have emerged as powerful tools for translational cancer research and preclinical drug screening. However, the limited availability of lung carcinoid cells, because of the small size of post-surgical samples, together with the low engraftment rate in mice, due to the slow growth of these neoplasms, hampered the employment of these procedures for TCs and ACs [12].

In the last decades, zebrafish (*Danio rerio*) has emerged among vertebrates as a powerful alternative model for the preclinical study of several human diseases, including cancer. Thanks to its intrinsic peculiarities, such as high fecundity, external fertilization, transparency, rapid embryo to larval transition, embryo permeability to small molecules, together with easy genetic manipulation and low maintenance cost, zebrafish has become a crucial and essential tool in biomedical research. In this context, we have developed a zebrafish xenograft platform based on the injection of human low-grade NETs (pituitary adenomas and medullary thyroid cancer) in transgenic *Tg(fli1a:EGFP)^y1^* zebrafish embryos at 48 h post-fertilization (hpf) [13,14,15,16,17,18], that express EGFP (enhanced green fluorescent protein) under control of the specific endothelial marker *fli1a*. This allows the visualization of the entire vascular tree in vivo. The injection of human tumor cells stimulates the migration and growth of sprouting vessels toward the implant within 24 to 72 h post-implantation (hpi), with the possibility to analyze the tumor-induced angiogenesis. At this stage, zebrafish embryos do not have a fully developed immune system and no graft rejection occurs. The use of fluorescent-labeled tumor cells allows the investigation of their metastatic behavior in zebrafish in real time. It is worthy of note that tumor xenograft in zebrafish embryos requires a small number of tumor cells, allowing the possibility to perform PDXs with post-surgical samples [19]. Moreover, taking advantage of the permeability of zebrafish embryos to small molecules, dissolved in their culture media, it is easy to study the anti-angiogenic activity of several drugs, in particular tyrosine kinase inhibitors. The rapidity of this procedure (within 5 days) makes this tool very useful to perform preclinical drug screening [17].

The aims of the present work were to develop a novel in vivo model based on the xenotransplantation of lung TC and AC cell lines and primary cell cultures in *Tg(fli1a:EGFP)^y1^* zebrafish embryos and to adopt this model for cancer drug testing. Here, we evaluated the anti-tumor activity of sulfatinib, a tyrosine kinase inhibitor, on lung carcinoid tumor cells implanted in zebrafish embryos.

## 2. Results

### 2.1. Tumorigenic Potential of Lung TC and AC Cell Lines and Primary Cell Cultures Implanted in Zebrafish Embryos

We assessed in vivo the ability of implanted lung TC and AC cells to stimulate angiogenesis in zebrafish embryos as early as 24 hpi and to migrate far from the transplantation site, focusing on the tail region at 48 hpi. Red-stained TC (NCI-H835, UMC-11, and NCI-H727) and AC (NCI-H720) cell lines, were implanted in the subperidermal space, between the periderm and the yolk syncytial layer, close to the sub-intestinal vein (SIV) plexus of 48 hpf *Tg(fli1a:EGFP)^y1^* embryos. Starting from 24 hpi, all injected cell lines induced a relevant pro-angiogenic response, leading to the formation of new endothelial structures, which sprouted from the SIV plexus and common cardinal vein (CCV) toward the tumor (Figure 1c–r), while we did not observe alterations of the normal vascular developmental pattern in the control embryos (Figure 1a,b). Tumor-induced angiogenesis was significantly higher in embryos implanted with NCI-H835, UMC-11, and NCI-H727 cells than that observed in NCI-H720 implanted embryos at both 24 (*p* < 0.001) and 48 hpi (*p* < 0.01) (Figure 1s,t).

The analysis of tumor cell migration far from the injection site showed that the presence of circulating tumor cells progressively increased, especially in the tail, within 48 hpi for all cell lines (Figure 2). In particular, the number of tumor cells along the tail resulted significantly higher in NCI-H720 implanted embryos than that observed in all TC-implanted embryos (*p* < 0.001) at 48 hpi (Figure 2m).

The zebrafish xenograft model was also effective for PDXs, allowing the possibility to implant a small number of cells with a highly successful implantation rate. As above described for immortalized cell lines, patient-derived lung carcinoid cells, derived from four patients, were red-stained and successfully implanted in the subperidermal cavity of 48 hpf *Tg(fli1a:EGFP)^y1^* zebrafish embryos. All four PDXs of lung carcinoid stimulated angiogenesis from 24 hpi, while in two out of four cases an invasive behavior has been observed with tumor cells in the tail of the embryo at 48 hpi. Images from one representative PDX in zebrafish are reported in Figure 3.

In order to better characterize the tumor-induced angiogenesis process in vivo and in real time, we performed time-lapse movies by the mean of selective plane illumination microscopy (SPIM) on implanted zebrafish embryos. We acquired 24 h lasting time-lapse movies of *Tg(fli1a:EGFP)^y1^* embryos implanted with NCI-H835, UMC-11, NCI-H727, and NCI-H720 tumor cells. Zebrafish embryos were followed for 24 h, repeating the acquisition every 10 min. Already in the first few frames, we observed endothelial structures sprouting from the SIV plexus that explored the tumor microenvironment, elongating toward the engraftment. These endothelial structures progressively converted into vessels that formed a complex vascular network around the tumor mass. The pro-angiogenic activity resulted in more relevance after the implantation of TC cells than after AC cell engraftment (Figure 4; Appendix A).

### 2.2. The Viability of Lung TC and AC Cells after the Engraftment in Zebrafish Embryos

We evaluated the proliferation rate of NCI-H835, UMC-11, NCI-H727, and NCI-H720 grafted cells in *Tg(fli1a:EGFP)^y1^* embryos through an immunofluorescence assay with a human Ki-67 antibody. This antibody is specific for human Ki-67; indeed no cross-reactivity occurs with zebrafish cells. All grafted cells (stained with a blue fluorescent dye) and implanted proliferating cells (positive for Ki-67 with red fluorescence) were visualized at fluorescence microscopy using two different color channels. After 48 hpi, we observed a high percentage of Ki-67-positive tumor cells for all injected cell lines, highlighting that tumor cells were actively proliferating even after the implantation in zebrafish (Figure 5).

### 2.3. Lung Carcinoid Xenograft in Zebrafish: A Platform for Drug Screening

In order to evaluate the potential role of this zebrafish platform for drug screening in lung carcinoids, we evaluated the effect of sulfatinib on tumor-induced angiogenesis and tumor cell invasiveness of NCI-H835, UMC-11, NCI-H727, and NCI-H720 cells implanted in *Tg(fli1a:EGFP)^y1^* zebrafish embryos. Sulfatinib is a small molecule tyrosine kinase inhibitor that targets tumor angiogenesis and immune modulation, inhibiting vascular endothelial growth factor receptors (VEGFRs) 1, 2, and 3, the fibroblast growth factor receptor type 1 (FGFR1), and colony-stimulating factor 1 receptor (CSF-1R) [20,21]. Due to the permeability of embryonic tissue to small molecules, sulfatinib was directly dissolved into the fish medium of 48 hpf correctly grafted embryos. Two different concentrations were tested, 0.25 and 2.5 µM, on the basis of preliminary pharmacological experiments on *Tg(fli1a:EGFP)^y1^* embryos without tumor xenograft, to detect the toxicity range of the drug, limiting the presence of morphological abnormalities. As controls we considered embryos treated with DMSO, the vehicle in which sulfatinib was dissolved.

Sulfatinib administration induced a dose-dependent inhibition of tumor-induced-angiogenesis, ranging between 30 and 50% in *Tg(fli1a:EGFP)^y1^* embryos injected with all lung carcinoid cell lines, after just 24 h of incubation. In embryos implanted with NCI-H720 and NCI-H727, an inhibitory effect on tumor-induced angiogenesis was observed only with the highest tested concentration (Figure 6).

After 2 days of treatment with sulfatinib, the spread of tumor cells outside the yolk sac was detected by fluorescence microscopy along the tail of *Tg(fli1a:EGFP)^y1^* implanted embryos. Sulfatinib did not significantly affect the number of circulating tumor cells in embryos injected with all tumor cell lines compared to DMSO-treated embryos, adopted as control (Figure 7).

## 3. Discussion

In patients with advanced lung carcinoid tumors, the therapeutic strategy is not curative and is directed at controlling symptoms from the tumor burden or hormonal production and slowing tumor growth [4,5,6,7,8]. The rarity of these tumors together with a restricted number of in vitro and in vivo models have hindered scientific progress compared with other cancers. Therefore, the development of novel in vivo models may shed new light on the mechanisms of tumor progression and therapeutic approaches.

Tumor xenografts, through the implantation of immortalized cell lines or primary cancer cells surgically resected from patients into an immunodeficient animal, are crucial tools for investigating the mechanisms involved in tumor progression and for drug screening [22,23,24]. The mouse xenograft model is considered the gold standard, but its use shows different drawbacks, such as a large number of cells necessary for tumor development, difficulties to generate tumor xenograft able to metastasize, long time to develop a visible tumor implant, immunosuppressed mice to avoid transplant rejection being more susceptible to infection, and high costs of maintenance [25,26].

Zebrafish has emerged as a worthwhile in vivo model organism in developmental biology and in the last years, it has been widely applied in cancer research. Zebrafish tumor xenograft platforms have been successfully developed to investigate different hallmarks of cancer, such as angiogenesis, cancer cell spread, and metastasis formation [19,27,28,29]. In addition, there are several studies that used the zebrafish xenograft model in the process of drug discovery and testing [17,30,31]. Notably, zebrafish xenografts could be considered a valid alternative to mouse in complying with ethical standards. Indeed, using zebrafish within 5 days post-fertilization should not fall under animal welfare regulation and it is in line with the “3R” (replacement, reduction, refinement) principle with the replacement of animals by alternative methods.

In the present work, we generated a zebrafish transplantable model for lung carcinoids, evaluating in vivo the ability of implanted lung TC and AC cells to stimulate angiogenesis and to migrate far from the transplantation site as early as 24 and 48 hpi. We took advantage of our zebrafish in vivo platform, based on xenotransplantation of NET cells in *Tg(fli1a:EGFP)^y1^* embryos at 48 hpf, expressing EGFP under the control of the endothelial-specific gene promoter *fli1a* [13,14,16].

We implanted several lung TC (NCI-H835, UMC-11, and NCI-H727) and AC (NCI-H720) cell lines, in order to evaluate and compare their tumorigenic potential in this in vivo platform. A strong tumor-induced angiogenesis was observed in embryos implanted with TC cell lines. Interestingly, it has been reported that well-differentiated NETs are more vascularized as compared to poorly differentiated neuroendocrine carcinomas [32,33,34,35]. This finding represents an interesting paradox, the so-called neuroendocrine paradox, where the most vascularized NETs are the most differentiated, but the high intratumoral microvascular density does not correlate with a high aggressiveness, in contrast to what is described for other cancers [36]. The neuroendocrine paradox is mainly related to gastroenteropancreatic NETs, but it has been also reported in lung NETs, where TCs are highly vascularized neoplasms with indolent behavior and good prognosis [37,38]. In addition, in our zebrafish model, AC cells (NCI-H720) displayed a lower pro-angiogenic potential and a higher ability to spread along the embryo body far from the injection site, compared to implanted TC cell lines. These results are in line with previous findings, showing that AC metastasized more frequently than TC [39], and strongly suggest that our zebrafish model faithfully recapitulates the tumor behavior of lung carcinoids in humans.

In order to follow in vivo and in real time the tumor-induced angiogenesis, we imaged the tumor xenograft in zebrafish with SPIM. This optical-sectioning imaging technique proved effective as a method for 3D time-lapse imaging of biological processes, such as vasculature development over long periods [40]. In SPIM, the selective illumination of the whole and only focal plane of the detection objective together with the parallel detection performed with a camera enable fast, volumetric imaging with high spatial resolution and high contrast, with a lower dose of light than standard microscopy technology such as confocal microscopy. Ultimately, SPIM reduces the effect of photobleaching and phototoxicity on the sample, which represents a relevant advantage for long-term time-lapse imaging. SPIM has been recently applied to visualize and track the behavior of metastatic cells in a breast and leukemia xenograft zebrafish model [41]. In the present study, for the first time, SPIM has been used to follow in real time the formation of tumor-induced vessels in our zebrafish/xenograft model. We acquired 24 h time-lapse movies, but by setting up the correct conditions for longer acquisitions, it would be possible to perform experiments for 48 h and longer. This would pave the way for the full description of the dynamics of tumor vasculature formation, with limited sufferance for the embryos. In addition, this technique provided a more detailed characterization of tumor-induced angiogenesis in zebrafish embryos implanted with lung carcinoid tumor cells with respect to that observed with the static fluorescence imaging, allowing a qualitative comparison between the angiogenic sprouting dynamics in TC and AC implanted embryos. Through SPIM, we confirmed the higher pro-angiogenic potential of TC cells compared to AC cells in zebrafish. The employment of SPIM may be useful to better characterize the formation of pathological vessels induced by neoplasms, providing potential clinical information. In particular, together with pharmacological strategies aimed at starving tumor cells, another relevant and complementary approach in cancer treatment is represented by tumor vascular normalization [42]. In tumors, uncontrolled angiogenesis is associated with an altered vascular maturation process. Structural abnormalities contribute to the dysfunction of tumor vessels, characterized by hypoperfusion, hyperpermeability, and severe hypoxia, resulting in limiting the delivery of cytotoxic agents to solid tumors. Indeed, the combination of antiangiogenic therapy with systemic chemotherapy often represents an effective strategy [43,44,45]. However, the window of vessel normalization is transient and hard to capture. The coupling of SPIM with microangiography in implanted zebrafish embryos could be particularly useful to study the vascular permeability in physiological and pathological conditions, as well as to detect the time window of the vascular normalization after antiangiogenic therapy. The identification of this extremely narrow window is a key factor in combination therapy to better define the optimal timing for drug administration.

The xenograft procedure in zebrafish requires the graft of a small number of tumor cells. This allows to implant tumor tissue directly from patients after surgical procedures. In zebrafish embryos, PDX of lung carcinoids showed pro-angiogenic and invasive behaviors within only 48 hpi and with a highly successful implantation rate. Currently, no data about lung carcinoid PDX models have been published. This model might be exploited to investigate molecular events involved in tumor-induced angiogenesis and migration in lung carcinoids, as well as to predict drug sensitivity in patients, opening a new scenario for the identification of the most appropriate and personalized treatment.

Here, we presented the reliability and potentialities of our zebrafish/xenograft platform for testing the effects of tyrosine kinase inhibitors on tumor-induced angiogenesis and migration of human TC and AC cells, implanted in *Tg(fli1a:EGFP)^y1^* zebrafish embryos. As a representative molecule of this class of drugs, we selected sulfatinib. Sulfatinib (also known as surufatinib) is a small molecule tyrosine kinase inhibitor targeting VEGFR 1, 2, and 3, FGFR1, and CSF-1R [20,21]. This drug showed promising anti-tumor activity with an acceptable and manageable safety profile in treating advanced pancreatic and extra-pancreatic neuroendocrine tumors [46,47]. In 2020, sulfatinib was approved as a monotherapy for unresectable locally advanced or metastatic, progressive non-functioning, well-differentiated (G1 or G2) extra-pancreatic neuroendocrine tumors in China. Moreover, an open-label phase II study evaluated the efficacy and tolerability of sulfatinib in patients with locally advanced or metastatic medullary thyroid cancer, showing that this therapy was well tolerated and the majority of patients achieved disease control [48]. Thanks to the permeability of zebrafish embryos in small molecules, sulfatinib was dissolved into the fish medium of 48 hpf grafted embryos. Sulfatinib significantly inhibited tumor-induced angiogenesis, ranging between 30 and 50%, in *Tg(fli1a:EGFP)^y1^* embryos implanted with all lung carcinoid cell lines, after only 24 h of treatment. This effect was more evident in zebrafish embryos with xenografts of TC cells (NCI-H835 and UMC-11). While sulfatinib showed no effect on the metastatic activity of all implanted carcinoid cells.

It has been reported that, in response to anti-VEGF therapies, some tumors can increase FGF secretion to stimulate endothelial cell proliferation, promote tumor angiogenesis, and bypass VEGF signaling pathways. Therefore, multi-target kinase inhibitors, such as sulfatinib, may represent a promising tool in treating NETs [49,50]. Targeting multiple kinases to simultaneously block VEGFR-, FGFR-, and CSF-1R-mediated pathways may be a more effective method of preventing tumor angiogenesis and tumor immune evasion, representing an attractive anti-cancer therapy approach [51,52,53,54]. So far, no data have been published on the anti-tumor activity of sulfatinib in patients with advanced lung carcinoids. Therefore, taking these considerations together with our results in the lung carcinoid zebrafish/xenograft model, we speculate that sulfatinib may represent a promising pharmacological approach for advanced lung carcinoids, targeting tumor-induced angiogenesis.

In conclusion, our zebrafish/xenograft model appears to be a cheap and fast platform for studying tumor-induced angiogenesis and tumor cell dissemination of lung carcinoids and to perform pharmacological testing. In addition, the rapidity of this procedure and the possibility to perform PDXs open a promising scenario for the identification of personalized therapies in patients with advanced lung carcinoid tumors.

## 4. Materials and Methods

### 4.1. Cell Lines and Reagents

Several human lung carcinoid tumor cell lines (NCI-H727, UMC-11, NCI-H835, and NCI-H720) were provided by ATTC. Cells were maintained at 37 °C in 5% CO_2_ and cultured in T75 flasks filled with 10 mL of RPMI medium (EuroClone™, Milan, Italy). Media was supplemented with 10% heat-activated fetal bovine serum (FBS) (EuroClone™, Milan, Italy) and 10^5^ U·L^−1^ penicillin/streptomycin (EuroClone™, Milan, Italy). Cells were harvested by trypsinization (Trypsin 0.05% and EDTA 0.02%) (Sigma-Aldrich^®^ Merck KGaA, Darmstadt, Germany), resuspended in complete medium, then counted through a Leica DM6000 B optical microscope (Leica Microsystems, Wetzlar, Germany) using a standard Beckman Coulter Z2 hemocytometer (Beckman Coulter, Brea, CA, USA) before plating. Cells used in all experiments were below 5 passages.

### 4.2. Primary Culture

Primary cell cultures were isolated from post-surgical samples of patients with pulmonary TC or AC, obtained after patients’ informed consent. Tumor cells were purified and plated as previously described [19]. Briefly, tumor cells were purified using anti-fibroblast microbeads (Miltenyi Biotec, Bergisch Gladbach, Germany). Viable cells were plated in D-valine Minimum Essential Medium (MEM, PromoCell, Heidelberg, Germany) to suppress the proliferation of any remaining fibroblasts, supplemented with 10% fetal bovine serum (FBS, Gibco-Thermo Scientific, Milano, Italy), 2 mM glutamine, and 1% penicillin/streptomycin (Lonza, Cologne, Germany) at 37 °C in a humidified atmosphere of 5% CO_2_ and 95% air. Procollagen immunofluorescent staining was performed to verify the absence of fibroblast contamination. Cell viability was assessed by trypan blue staining before the injection, and it was higher than 90%.

### 4.3. Zebrafish Line and Maintenance

Embryo and adult zebrafish were raised and maintained according to Italian (D.Lgs 26/2014) and European laws (2010/63/EU and 86/609/EEC). Embryos were collected by natural spawning and staged according to morphological criteria [55]. Starting from 24 hpf, embryos were cultured in fish water (0.1 g/L NaHCO_3_, 0.1 g/L Instant Ocean, 0.192 g/L CaSO_4_•2H_2_O) containing 0.003% PTU (1-phenyl-2-thiourea; Sigma-Aldrich, Saint Louis, Mo) to prevent pigmentation and 0.01% methylene blue to prevent fungal growth. All experiments were performed on *Tg(fli1a:EGFP)^y1^* transgenic embryos [56]. At 24 hpf, after removing the chorion, embryos were incubated for a further 24 h at 28 °C.

### 4.4. Zebrafish Tumor Xenograft

The 48 hpf *Tg(fli1a:EGFP)^y1^* embryos were anesthetized with 0.016% tricaine (Ethyl 3-aminobenzoatemethanesulfonate salt, Sigma-Aldrich^®^ Merck KGaA, Darmstadt, Germany) and placed on agarose-modified Petri dish, where they were oriented with the yolk on a flank and implanted with human lung carcinoid tumor cell lines and primary cell cultures as previously described [13,19]. Briefly, tumor cells were labeled with red (CellTracker^TM^ CM-DiI Dye, Invitrogen™ Thermo Fisher Scientific, Waltham, MA, USA) or blue (CellTracker^TM^ Blue CMAC, Invitrogen™ Thermo Fisher Scientific, Waltham, MA, USA) fluorescent viable dye following the manufacturer’s instructions, resuspended with PBS, and grafted (about 100–1000 cells per embryo) into the sub-peridermal space of *Tg(fli1a: EGFP)^y1^* embryos, close to the SIV plexus. As a control of the implantation, we considered embryos injected with only PBS. Injections were performed by a micro-injector FemtoJet (Eppendorf, Hamburg, Germany), equipped with a micromanipulator InjectMan NI 2 (Eppendorf, Hamburg, Germany). All implanted embryos were raised at 32 °C, a compromise temperature between 28 °C (the optimal temperature for zebrafish maintenance) and 37 °C (the optimal temperature for mammalian cell growth and metabolism).

### 4.5. In Vivo Zebrafish Assays for Tumor-Induced Angiogenesis and Tumor Cell Migration

Starting from 24 hpi, the pro-angiogenic and migratory responses of injected tumor cells were monitored in vivo.

To measure the arbitrary unit of tumor-induced angiogenesis, we calculated in each imaged embryo the EGFP area corresponding to endothelial structures that sprouted from the SIV plexus by using the open-source Fiji software (ImageJ2, NIH, Bethesda, Rockville, MD, USA, available at https://imagej.net/software/fiji/ accessed on 1 July 2022). Data were normalized against the mean of the lowest value, arbitrary set to 1.0.

The presence of tumor cell clusters far from the injection site was detected by fluorescence microscopy and quantified by the “Analyze Particle” tool of Fiji software. In particular, fluorescent cells along the tail were quantified at 48 hpi. Data were normalized against the mean of the lowest values, arbitrarily set to 1.0.

For each tumor cell type, assays were performed 3 times, considering about 20 embryos in each experimental group.

All images were taken at 0, 24, and 48 hpi with a Leica M205 FA stereomicroscope equipped with a Leica DFC 450 C digital camera using the LAS V4.2 software (Leica Microsystems, Wetzlar, Germany).

### 4.6. Selective Plane Illumination Microscopy (SPIM)

The samples were imaged in a dual illumination Light Sheet Fluorescence Microscope consisting of three identical water-dipping objectives (UMPLFLN 10×/0.3, Olympus, Tokyo, Japan). One of the three objectives was used for detection. Two objectives, perpendicular to the detection objective, were used for illumination. A 488 nm laser (RGB Photonics) was used for GFP fluorescence excitation. The laser beam was split 50/50 to illuminate the sample consecutively from opposing directions, in order to increase the laser penetration. The beam was sent to a resonant galvanometric mirror (200 Hz, Thorlabs, Newton, NJ, USA), which pivots the light sheet and reduces shadowing effects in the excitation paths due to absorption in the specimen. Light sheets were generated with cylindrical lenses (f = 50 mm) after being expanded with telescopes and were projected by the illumination objectives onto the focal plane of the detection lens. The sample was dipped in the medium-filled imaging chamber from the top and was translated along the *z*-axis with a linear stage (M-404.1PD, Physik Instrumente). The focal plane of the detection objective was imaged with an sCMOS camera (Neo, Andor, Belfast, UK). The camera captured images of a field of view of ∼1 mm^2^ at 30–100 frames per second, generating volumetric stacks of 100 planes, equally spaced by 10 um. Before acquisitions, *Tg(fli1a:EGFP)^y1^* embryos implanted with NCI-H835, UMC-11, NCI-H727, and NCI-H720 were anesthetized with 0.016% tricaine and then mounted in FEP (fluorinated ethylene propylene) tubes, as previously described [57,58]. In order to improve the zebrafish resistance during the acquisition, a temperature-controlled chamber was developed. A Peltier cell was installed below the imaging chamber and used with a reversed current in order to keep the temperature constant at 32°, in a feedback loop with a temperature sensor immersed in the liquid. Implanted zebrafish embryos were kept under observation for 24 h, repeating the acquisition every 10 min.

### 4.7. Immunofluorescence Assays

At 48 hpi, embryos were fixed with a 4% PFA solution, and whole-mount immunofluorescence assays were performed as previously described [59] with the specific monoclonal antibody anti-human Ki-67 (Dako, Agilent Technologies, Santa Clara, CA, USA) and Alexa fluor 555 goat anti-mouse (Molecular Probes, Eugene, OR), for the evaluation of proliferating tumor cells. Briefly, embryos were washed in PBS with 0.1% Tween20, permeabilized for 20 min in cold acetone, incubated in NH_4_Cl solution for 30 min at room temperature, blocked for 2 h at room temperature in blocking solution (5% bovine serum albumin (BSA) in PBS- 0.1% Tween20) and incubated overnight at 4 °C in anti-human Ki-67 antibody 1:200 in blocking solution. The next day, embryos were washed every hour with blocking solution at room temperature until the incubation overnight at 4 °C in Alexa fluor 555 goat anti-mouse 1:500 in blocking solution. On the last day, embryos were washed in 0.1% Tween20/PBS at room temperature. All images were taken with a Leica M205 FA stereomicroscope equipped with a Leica DFC 450 C digital camera using the LAS V4.2 software (Leica Microsystems, Wetzlar, Germany).

### 4.8. Pharmacological Treatments

After the implantation of tumor cells, zebrafish embryos were treated up to 48 h with sulfatinib, directly dissolved into fish water. Sulfatinib was provided by MedChemExpress LLC (Monmouth Junction, NJ, USA). Stock solution (10 mM) was made in 100% dimethyl sulfoxide (DMSO). As untreated controls, we considered injected embryos incubated in the fish water and the vehicle in which the experimental substance was dissolved (DMSO). Assays were performed 3 times, considering about 20 embryos in each experimental group. Tumor-induced angiogenesis and tumor cell migration analyses were performed as described above. Data were normalized against the mean of the control (DMSO), arbitrarily set to 1.0.

### 4.9. Statistical Analysis

All experiments were performed at least three times, considering at least 20 embryos in each experimental group. Statistical differences among groups were evaluated by ANOVA test together with a post hoc Tukey’s multiple comparison test. A *p*-value < 0.05 was considered statistically significant. The values reported in the graphs represent the mean ± standard error of the mean (S.E.M). For statistical analysis, GraphPad Prism 5.0 was used (GraphPad Software Inc., La Jolla, CA, USA).

## Figures and Tables

**Figure 1 ijms-23-08126-f001:**
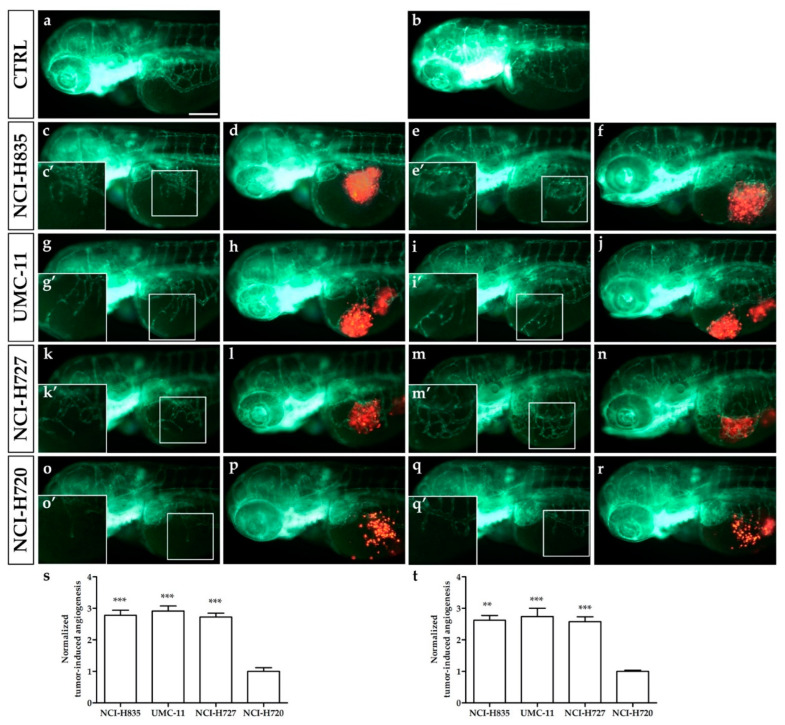
Engraftment of lung carcinoid cells in zebrafish embryos. Representative epifluorescence images of *Tg(fli1a:EGFP)^y1^* embryos injected with cell resuspension solution (control: **a**,**b**) and implanted with red-stained NCI-H835, UMC-11, NCI-H727, and NCI-H720 cells (**c**–**r**). Embryos were imaged at 24 (**a**,**c**,**d**,**g**,**h**,**k**,**l**,**o**,**p**) and 48 (**b**,**e**,**f**,**i**,**j**,**m**,**n**,**q**,**r**) hpi. The red channel was omitted in panels (**c**,**e**,**g**,**i**,**k**,**m**,**o**,**q**) to facilitate the observation of tumor-induced angiogenesis (green). (**c**′,**e**′,**g**′,**i**′,**k**′,**m**′,**o**′,**q**′) are the digital magnifications of white-boxed regions. The graphs showed the quantification of tumor-induced angiogenesis in embryos implanted with lung carcinoid cells after 24 (**s**) and 48 (**t**) hpi. NCI-H720 values have been set to 1.0. Graphed values represent the mean ± S.E.M. ** *p* < 0.01 vs. NCI-H720; *** *p* < 0.001 vs. NCI-H720. All implanted cells stimulated endothelial sprouting from the SIV plexus within 24 hpi. Embryos are shown anterior to the left. Scale bar: 100 µm.

**Figure 2 ijms-23-08126-f002:**
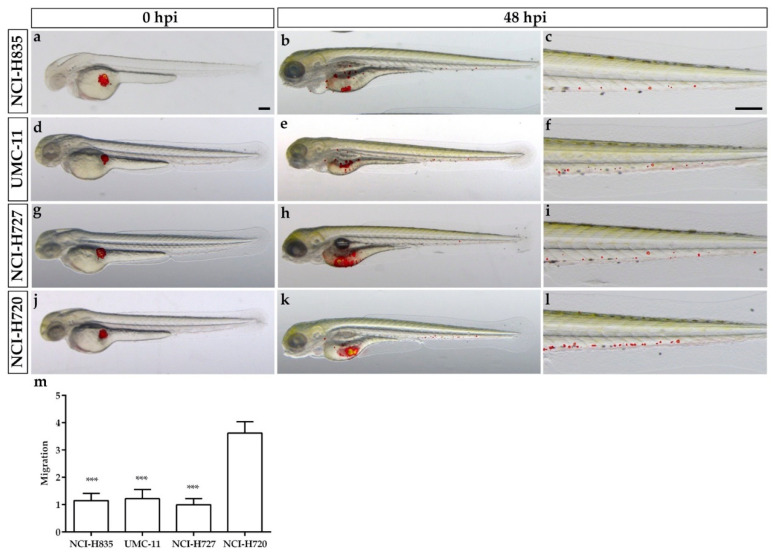
Lung carcinoid cell invasiveness in grafted zebrafish embryos. Overlay of representative fluorescent and bright field images of embryos grafted with red-stained NCI-H835, UMC-11, NCI-H727, and NCI-H720 cells at 0 (**a**,**d**,**g**,**j**) and 48 hpi (**b**,**e**,**h**,**k**), respectively. For each injected cell line, the tail, particularly at 48 hpi, was imaged (**c**,**f**,**i**,**l**). Images showed the spread of carcinoid cells throughout the embryo body. The graph showed the quantification of tumor cell migration in the tail of embryos implanted with lung carcinoid cells after 48 hpi (**m**)**.** As an arbitrary unit of migration, we considered the number of fluorescent particles in the tail. NCI-H727 values have been set to 1.0. Graphed values represent the mean ± S.E.M. *** *p* < 0.001 vs. NCI-H720. Embryos are shown anterior to the left. Scale bar: 100 µm.

**Figure 3 ijms-23-08126-f003:**
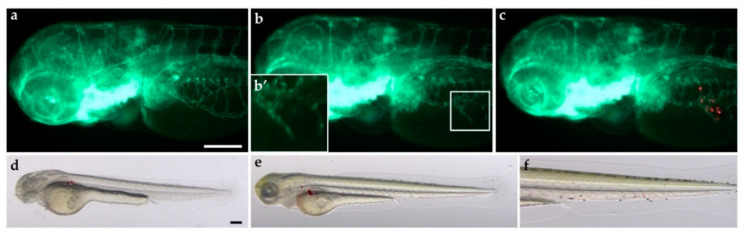
Tumorigenic potential of a lung carcinoid PDX in zebrafish embryos. Red-stained cells, obtained from a patient surgical resection, were used to perform PDX in 48 hpf *Tg(fli1a:EGFP)^y1^* zebrafish embryos. Epifluorescence images at 24 hpi of cell resuspension solution injected embryos (control: **a**) and patient-derived lung carcinoid (red) xenografted embryos (**b**,**c**). The red channel was omitted in panels (**b**,**b**′) to facilitate the observation of tumor-induced angiogenesis (green); **b′** is the digital magnification of the white-boxed region. Lung carcinoid PDX induced the formation of endothelial structures (green) sprouting from the SIV within 24 hpi. Overlay of representative fluorescent and bright field images of grafted embryos at 0 (**d**) and 48 hpi (**e**,**f**) showed the spread of tumor cells throughout the embryo body. The tail, particularly at 48 hpi, was imaged (**f**). Embryos are shown anterior to the left. Scale bar: 100 µm.

**Figure 4 ijms-23-08126-f004:**
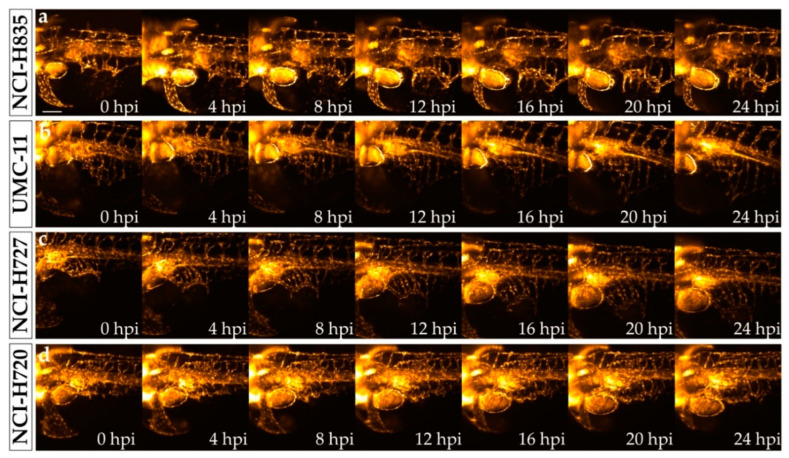
Time-lapse imaging of lung carcinoid cell grafted embryos performed with selective plane illumination microscopy. Representative maximum intensity projections of volumetric stacks acquired of *Tg(fli1a:EGFP)^y1^* zebrafish embryos at the level of the graft regions captured every 4 h until 24 hpi. Embryos were implanted with NCI-H835 (**a**), UMC-11 (**b**), NCI-H727 (**c**), and NCI-H720 (**d**) cells. Embryos are shown anterior to the left. Scale bar: 100 μm.

**Figure 5 ijms-23-08126-f005:**
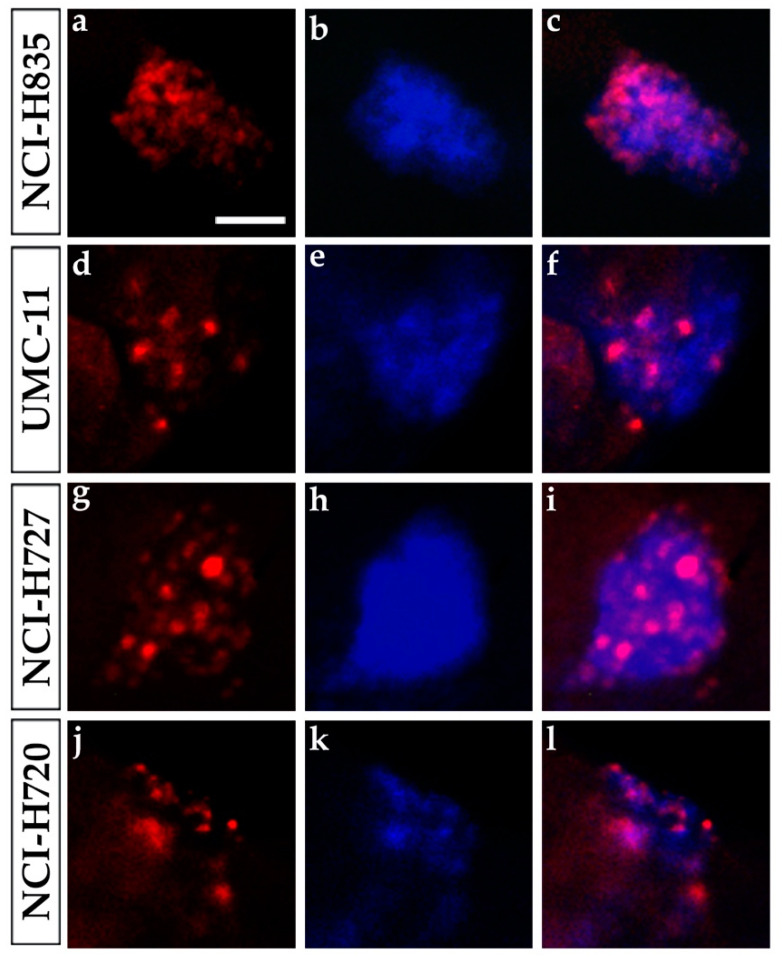
Ki-67 immunostaining of lung carcinoid grafted embryos. Representative images of 48 hpi *Tg(fli1a:EGFP)^y1^* zebrafish embryos at the level of the graft regions after immunofluorescence assay to detect Ki-67 localization (red). Embryos were implanted with NCI-H835 (**a**–**c**), UMC-11 (**d**–**f**), NCI-H727 (**g**–**i**), and NCI-H720 (**j**–**l**) cells. Injected lung carcinoid cells were previously blue-stained (**b**,**e**,**h**,**k**). The merge of red and blue channels (**c**,**f**,**i**,**l**) showed that Ki-67 staining is tumor cell specific. Scale bar: 50 µm.

**Figure 6 ijms-23-08126-f006:**
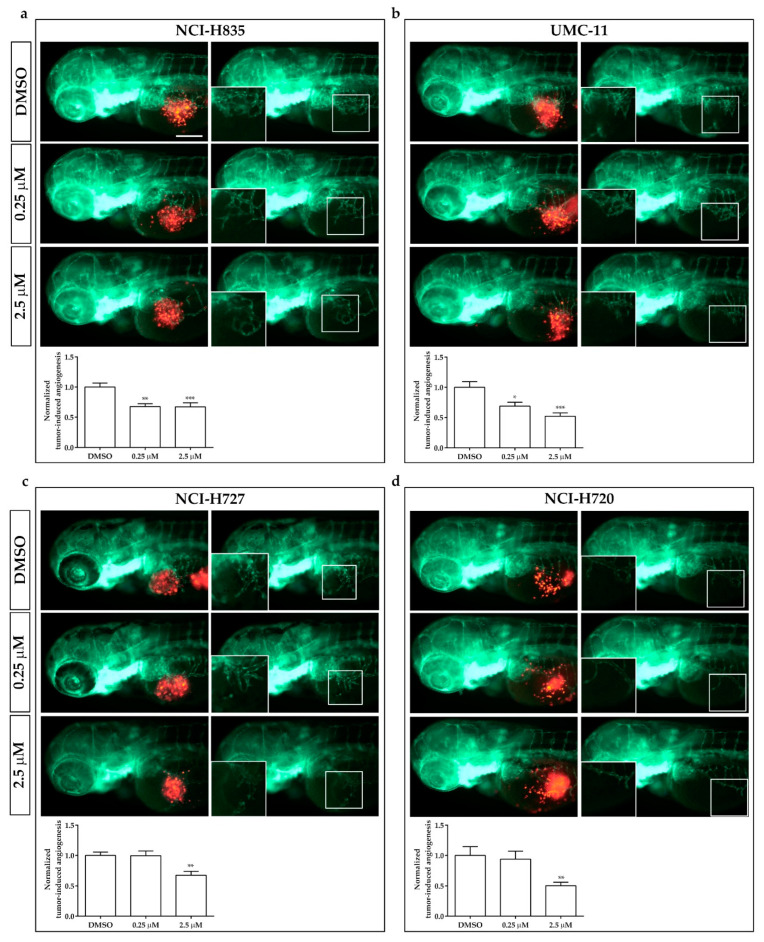
Effects of sulfatinib on tumor-induced angiogenesis in zebrafish embryos implanted with lung carcinoid cells. Representative epifluorescence images of *Tg(fli1a:EGFP)^y1^* zebrafish embryos at 24 hpi, implanted with NCI-H835 (**a**), UMC-11 (**b**), NCI-H727 (**c**), and NCI-H720 (**d**) cells and treated with DMSO (vehicle control) and sulfatinib (0.25 and 2.5 µM). The red channel, corresponding to lung carcinoid cells, was omitted in the second column of each panel to highlight the tumor-induced microvascular network. Digital magnifications of the graft regions are shown in white boxes. At the bottom of each panel, the graph showed the quantification of tumor-induced angiogenesis in embryos implanted with lung carcinoid cells after 24 h of treatment with DMSO and sulfatinib (0.25 and 2.5 µM). Control (DMSO) values have been set to 1.0. Graphed values represent the mean ± S.E.M. * *p* < 0.05 vs. DMSO; ** *p* < 0.01 vs. DMSO; *** *p* < 0.001 vs. DMSO. Embryos are shown anterior to the left. Scale bar: 100 µm.

**Figure 7 ijms-23-08126-f007:**
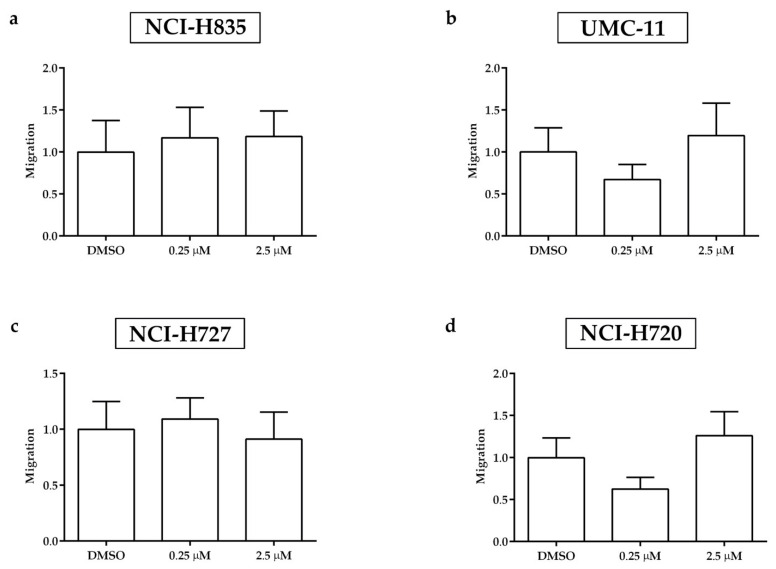
Effects of sulfatinib treatments on the invasiveness of lung carcinoid cells in grafted zebrafish embryos. Quantification of cell spread in the tail of embryos injected with NCI-H835 (**a**), UMC-11 (**b**), NCI-H727 (**c**), and NCI-H720 (**d**) cells at 48 hpi after 0.25 and 2.5 µM sulfatinib 48 h treatments. As an arbitrary unit of migration, we considered the number of fluorescent particles in the tail. DMSO values have been set to 1.0. Graphed values represent the mean ± S.E.M.

## Data Availability

The data presented in this study are available on request from the corresponding author.

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
