# Peer review of "Modeling Lung Carcinoids with Zebrafish Tumor Xenograft"

_ijms, 2022, doi:10.3390/ijms23158126_

Round 1

Reviewer 1 Report

MAJOR POINTS

This is an outstanding paper describing a novel approach to preclinical research in the field of Thoracic Neuroendocrine tumors. The author describe  a novel in vivo model of lung carcinoids based on the xenograft of different cell lines of Typical and Atypical Carcinoid and patient-derived cell cultures and describe the potential role in in assess in vivo response to different drugs utilizing different pre/clinical parameters. The description, the methods and the iconography (pictures in the manuscript and videos in the supplementary files are extremely suggestive) and may have an extreme interest for the readers and may trace the road for further experimental experience. 

MINOR POINTS

No particular corrections are necessary, we only suggest to add to the references the ENETS and ESMO International guidelines for the treatment of thoracic neuroendocrine tumors as a base for the comprension of the importance of the need of new therapeutic strategies to be tested in these field since the possibility of cure are still limited in the metastic setting  

Author Response

We thank the reviewer for the suggestion. We added both references in the introduction (ref n. 10 and 11):

  1. Baudin, E.; Caplin, M.; Garcia-Carbonero, R.; Fazio, N.; Ferolla, P.; Filosso, P.L.; Frilling, A.; de Herder, W.W.; Horsch, D.; Knigge, U.; Korse, C.M.; Lim, E.; Lombard-Bohas, C.; Pavel, M.; Scoazec, J.Y.; Sundin, A.; Berruti, A.; clinicalguidelines@esmo.org, E.G.C.E.a. Lung and thymic carcinoids: ESMO Clinical Practice Guidelines for diagnosis, treatment and follow-up. Ann Oncol 2021, 32, (4), 439-451. 10.1016/j.annonc.2021.01.003.
  2. Caplin, M.E.; Baudin, E.; Ferolla, P.; Filosso, P.; Garcia-Yuste, M.; Lim, E.; Oberg, K.; Pelosi, G.; Perren, A.; Rossi, R.E.; Travis, W.D.; participants, E.c.c. Pulmonary neuroendocrine (carcinoid) tumors: European Neuroendocrine Tumor Society expert consensus and recommendations for best practice for typical and atypical pulmonary carcinoids. Ann Oncol 2015, 26, (8), 1604-20. 10.1093/annonc/mdv041.

Reviewer 2 Report

The manuscript "Modeling lung carcinoids with zebrafish tumor xenograft", written by Carra S, Gaudenzi G, Dicitore A, Cantone MC, Plebani A, Saronni D, Zappavigna S, Caraglia M, Candeo A, Bassi A, Persani L and Vitale G. presents experiments on zebrafish embryos as models for exploring angiogenesis and migration of cancer cells. The authors implanted several types of lung carcinoid tumor cells in fish embryo and followed their migration, proliferation and angiogenesis during first two days of life, by different types of microscopy. Beside tumor cell lines, zebrafish was also used as a model for patient derived xenografts. Angiogenesis and migration of tumor cells were also analyzed after treatment of zebrafish with sulfatinib, inhibitor of angiogenesis.

The manuscript is well written, data clearly presented, methods described in details.

Minor comments:

I would suggest to mention sulfatinib targets (kinases linked with angiogenesis) in Results.

line 161: sentence reorganization

line 275: in our zebrafish model,

Author Response

Minor comments:

I would suggest to mention sulfatinib targets (kinases linked with angiogenesis) in Results.

REPLY: We thank the reviewer for the comment about our paper. We mentioned sulfatinib targets in the Results.

line 161: sentence reorganization

REPLY: We rephrased the sentence according to the suggestion.

line 275: in our zebrafish model,

REPLY. We added the comma.

Reviewer 3 Report

The work received for review concerned modeling of lung carcinoids with zebrafish tumor xenograft. An interesting work that fits in with the important topic of the fight against cancer. It is extremely important to have the tools to test new anti-cancer therapies. However, the reviewer has a few questions and comments regarding the work.

1. Why the authors decided to check only one drug - sulfatinib. Wouldn't it be better to compare the results for 3 different therapeutics? Please also explain the choice of medication in more detail.

2. In Materials and Methods, all numbers of the reagent formulas should be subscript.

3. Please indicate the type, manufacturer of the mixoscope and haemocytometer.

4. Have you obtained the consent of the Bioethics Committee for the collection of primary tumor cells.

Author Response

  1. Why the authors decided to check only one drug - sulfatinib. Wouldn't it be better to compare the results for 3 different therapeutics? Please also explain the choice of medication in more detail.

REPLY. The aim of our work was to demonstrate the power of our zebrafish/xenograft model for lung carcinoids. Here, we showed the reliability and potentialities of this platform for drug screening, testing the effects of sulfatinib, as representative of small molecule tyrosine kinase inhibitors. The selection of sulfatinib is based on the promising anti-tumor activity reported in in 2 recent clinical trials enrolling patients with advanced pancreatic and extra-pancreatic neuroendocrine tumors (page 11). The comparison of different drugs for the treatment of lung carcinoids, taking advantage of our zebrafish/xenograft platform, will be the focus of another paper.  

  1. In Materials and Methods, all numbers of the reagent formulas should be subscript.

REPLY: We corrected the reagent formulas in M&M section.

  1. Please indicate the type, manufacturer of the mixoscope and haemocytometer.

REPLY. We indicated models and manufacturers of optical microscope and haemocytometer.

  1. Have you obtained the consent of the Bioethics Committee for the collection of primary tumor cells.

REPLY: In the M&M (page 12) and in the Informed Consent Statement section (page 15) we reported that “The Informed consent was obtained from all subjects for their tissue to be used in this study”. In the Institutional Review Board Statement section (page 15), we have specified: “… The study and related informed consent form were approved by the Ethics Committee of Istituto Auxologico Italiano (protocol code: 2020_01_28_04, date of approval: 28th January 2020)”.